# Use of a Machine Learning Method in Predicting Refraction after Cataract Surgery

**DOI:** 10.3390/jcm10051103

**Published:** 2021-03-06

**Authors:** Tomofusa Yamauchi, Hitoshi Tabuchi, Kosuke Takase, Hiroki Masumoto

**Affiliations:** 1Department of Ophthalmology, Tsukazaki Hospital, Himeji 671-1227, Japan; t.yamauchi@tsukazaki-eye.net (T.Y.); k.takase@tsukazaki-eye.net (K.T.); h.masumoto@tsukazaki-eye.net (H.M.); 2Department of Technology and Design Thinking for Medicine, Hiroshima University, Hiroshima 734-8511, Japan

**Keywords:** IOL power calculation, machine learning, gradient booting regression (GBR), neural network, support vector regression (SVR), random forest regression (RFR)

## Abstract

The present study aims to describe the use of machine learning (ML) in predicting the occurrence of postoperative refraction after cataract surgery and compares the accuracy of this method to conventional intraocular lens (IOL) power calculation formulas. In total, 3331 eyes from 2010 patients were assessed. The objects were divided into training data and test data. The constants for the IOL power calculation formulas and model training for ML were optimized using training data. Then, the occurrence of postoperative refraction was predicted using conventional formulas, or ML models were calculated using the test data. We evaluated the SRK/T formula, Haigis formula, Holladay 1 formula, Hoffer Q formula, and Barrett Universal II formula (BU-II); similar to ML methods, we assessed support vector regression (SVR), random forest regression (RFR), gradient boosting regression (GBR), and neural network (NN). Among the conventional formulas, BU-II had the lowest mean and median absolute error of prediction. Therefore, we compared the accuracy of our method with that of BU-II. The absolute errors of some ML methods were lower than those of BU-II. However, no statistically significant difference was observed. Thus, the accuracy of our method was not inferior to that of BU-II.

## 1. Introduction

With recent advances in cataract surgery technology, the importance of predicting postoperative refractive power has increased relatively more [1]. In addition, accurate refraction prediction is essential for the use of multifocal intraocular lenses, which have become widely used in recent years [2]. For these reasons, the need for accuracy in intraocular lens (IOL) power calculation is greater than ever. Although formulas are becoming more and more accurate, the highest possible accuracy is desired [3]. Satou et al. have recently reported a formula that uses detailed anatomical measurements of the anterior eye using anterior segment Optical Coherence Tomography (AS-OCT). Their formula shows high accuracy without being affected by the axial length of the eye [4]. In addition, several papers that used numerous intraocular lens (IOL) power calculation formulas in predicting postoperative refraction have been published, and recent reports have indicated that the Barrett Universal II formula has high accuracy [5,6,7]. On the other hand, Kane et al. recently reported that Kane’s formula has higher accuracy in studies with many cases [8].

The IOL power calculation formula is a regression equation used to predict postoperative refraction with preoperative parameters, such as axial length and corneal curvature, and an anterior chamber depth and powers and types of implanted intraocular lenses. [9]. The initial power formula is basically a prediction based on the anatomical features of the eye and optical calculations [10,11,12,13]. The aforementioned Barrett Universal II formula also uses this method [14,15]. However, with recent developments in computer science, the use of methods incorporating machine learning (ML) has been reported [16,17]. Therefore, the present study aimed to assess the accuracy of the available methods for predicting postoperative refraction using ML. We evaluated four ML methods, including support vector regression (SVR) [18], random forests regression (RFR) [19], gradient boosting regression (GBR) [20], and neural network (NN) [21]. This study aimed to create a more accurate model with a relatively small number of cases. A unique aspect of this investigation was the application of the predicted postoperative refraction using the conventional IOL power calculation formula as an explanatory variable in ML. We also examined whether this has higher accuracy than the original formula.

## 2. Methods

### 2.1. Study Design

The current research utilized a retrospective study design. Figure 1a shows the flow of prediction using the conventional IOL calculation formulas. The constants for the formulas were optimized using training data. Figure 1b depicts the flow of prediction using ML methods.

Figure 1 Summary of the training and test data.

In ML, we trained the model using training data, and the test data were then applied to predict postoperative refraction. The absolute values of refractive errors and the proportion of objects with absolute errors of refractions less than 0.5 D were evaluated.

### 2.2. Patients

Patients were consecutive cases (*n* = 3331) who underwent cataract surgery at the Tsukazaki Hospital between October 2017 and January 2019 and met the inclusion criteria. The inclusion criteria were eyes without ocular disease other than cataracts. For example, Keratoconus, or moderate or greater glaucoma (The MD value by Humphrey Field Analyzer < −6D was used as an exclusion criterion because this number is widely used in academic circles as the number that separates the middle and early stages of glaucoma [22].) or diabetic retinopathy were excluded. Since the refractive value from the subjective visual acuity test was used, glaucoma after the middle stage, which may have reduced central visual field function, was excluded because it may cause variation in subjective refractive values. Eyes with a postoperative corrected distance visual acuity less than 16/20 were excluded. The reason is that these patients may not have accurate ophthalmologic examinations due to old age, mental illness, or dementia. Moreover, eyes in which measurement using the IOLMaster 700 (Carl Zeiss, Oberkochen, Germany) was unsuccessful before or after surgery, were not included. We used a total of 2831 eyes for training for ML: 487 unilateral entries in 487 cases and 1172 binocular entries in 2344 eyes. For the evaluation of ML performance, 500 eyes per model were tested using only unilocular entries. Table 1 shows a summary of implanted IOLs.

For the test data, we randomly selected 500 YP2.2 implanted eyes from 500 patients. Then, we excluded the fellow eyes of the test data from objects. For the training data, we used 296 YP2.2 (KOWA), 260 SZ-1 (NIDEK), 193 W60R (Santen), 28 KS-SP (STAAR), 21 NS60YG (NIDEK), 125 SN60WF (Alcon), 208 SN6AT series (Alcon), 79 SN6AD (Alcon), 38 SV25T series (Alcon), 463 ZCB00V (J&J Medical), 463 TECNIS Multifocal series (ZMB00, ZLB00, and ZKB00; J&J Medical), and TECNIS symphony series (ZXR series and ZXV series; J&J medical). We also conducted subgroup analysis on the basis of axial length. Our previous study showed that mean axial length of Japanese was 23.6 mm [21]. Using this data as a reference, subgroup analysis was performed such that axial length of less than 22 mm was defined as the short-axis group (*n* = 10), that between 22 mm to less than 24 mm was defined as the middle-axis group (*n* = 301), and axial length of 24 mm or greater was defined as the long-axis group (*n* = 189).

### 2.3. Preoperative, Postoperative Examinations, and Surgical Measurements

Preoperatively, we assessed axial length, corneal curvature, anterior chamber depth, lens thickness, and white-to-white distance using IOLMaster 700.

Ten weeks after surgery, we measured the same items similar before surgery using IOLMaster 700. Distance visual acuity was measured at 5.0 m using the decimal visual acuity chart. For postoperative refraction, we used subjective spherical equivalent.

Six experienced surgeons performed cataract surgeries. 2.4-mm temporal corneal incision was made. Next, a 5.0-mm continuous curvilinear capsulorhexis was created.

### 2.4. IOL Power Calculation Formulas and Optimization of Constants

Regarding conventional IOL power calculation formulas, we evaluated the SRK/T formula [23], Haigis formula [9], Holladay 1 formula [10], Hoffer Q formula [11,12], and Barrett Universal II formula [13].

For the optimization of A constant (SRK/T formula), surgeon factor (Holladay 1 formula), and personalized Anterior Chamber Depth (pACD) (Hoffer Q formula), Excel (Microsoft, Albuquerque, NM, USA) was used, and the calculation was performed using training data. For the optimization of a0, a1, and a2 (Haigis formula), we used linear regression in accordance with the original method. Linear regression was performed using Python 3 (https://www.python.org/ (accessed on 25 February 2021) Python Software Foundation, Delaware, DE, USA) and Scikit-learn library (http://scikit-learn.org/stable/ (accessed on 25 February 2021) Free software machine learning library for the Python programming language) using training data. For the Barrett Universal II formula, the optimized A constant was used.

The predicted postoperative refractions of SRK/T formula, Haigis formula, Holladay 1 formula, and Hoffer Q formula were calculated using Excel. In detail, these published formulas were created using Excel functions and calculated by inputting measurements such as ocular axis length IOL power, and each optimized IOL constant. These values were calculated using the test data and optimized constants in accordance with the original methods. The predicted postoperative refractions of Barrett Universal II formula were calculated using a calculator on the website.

For the SRK/T formula, Holladay 1 formula, Hoffer Q formula, and Haigis formula, we used constants that optimized for each formula. In contrast, for the Barrett Universal II formula, A constants optimized for the SRK/T formula were used. Therefore, to minimize the error of power calculation using the Barrett Universal II formula, we calculated the average of the refraction errors in the training data for the Barrett Universal II formula; then, the average was subtracted from the calculated predicted refractions in the test data. Optimized constants for the IOL power calculation formula of the datasets are presented in the Table 2

### 2.5. Machine Learning

ML was conducted using a self-made program with Python 3. The Scikit-learn library (http://scikit-learn.org/stable/ (accessed on 25 February 2021) was used for SVR and RFR. For GBR, the XGboost library (https://github.com/dmlc/xgboost (accessed on 25 February 2021) was utilized. For NN, we used the TensorFlow library (https://github.com/tensorflow/tensorflow (accessed on 25 February 2021) as a backend and the Keras library (https://github.com/keras-team/keras (accessed on 25 February 2021) as a wrapper. The structure of the NN is depicted in the Figure 2.

Because random numbers are used for learning with NN, some fluctuations occurred in the prediction result. Therefore, for NN, learning was repeated 30 times and the average value was used.

The parameters to be used as explanatory variables were selected based on the GBR prediction accuracy in the training data. There are two reasons for using GBR for parameter selection. One is that the feature importance can be obtained by calculation, and the other is that the calculation speed is fast and therefore suitable for cross-validation. First, we selected a formula whose predicted values were used as explanatory variables. We selected the formule with the highest feature importance for prediction among the four conventional formulas (SRK/T formula, Holladay 1 formula, Hoffer Q formula, and Haigis formula). We excluded the predicted value of the Barrett Universal II formula from the candidates for explanatory variables because the details of the formula have not been published and also to avoid autoregressive behavior. Next, we selected the parameters considered for use as the explanatory variables of ML. The candidates for explanatory variables were age, axial length, corneal curvature, anterior chamber depth, lens thickness, white-to-white distance, IOL constants (optimized value), IOL power, and predicted refraction using the selected conventional formula. Gender was not considered because it did not contribute much to accuracy in our previous unpublished study. It showed that F-value, which means the feature importance, was the smallest in the gender of all parameters. (F-values: Axial length, 217; corneal curvature radius,123; lens power, 121; lens thickness, 24; white to white,23; ACD, 22; A constant,21; age, 17; gender,13) Prior to the use of these values, they were normalizes to obtain an average of 0 and a standard deviation of 1. We selected a combination of parameters that minimized the error via cross-validation in the training data. As a dependent variable, postoperative refraction was used. Hyperparameters in ML were optimized via cross-validation using grid-search with the training data.

### 2.6. Statistical Analysis

For statistical analysis, we used Python 3 and the SciPy library (https://www.scipy.org/ (accessed on 25 February 2021). Unpaired *t*-tests were utilized to compare average values of continuous variables (such as age or axial length) between training and test data. When comparing the accuracy of each IOL power calculation formula and ML method, we used the Shapiro–Wilk test to evaluate data normality and the Friedman test to determine whether there were differences between groups. Lastly, the paired *t*-test or the Wilcoxon signed-rank test was used, depending on the nature of the distribution. To compare the proportion of objects with refraction errors less than 0.5 D, chi-squared tests were utilized. We applied the Bonferroni correction for multiple comparisons when the *p*-value was less than 0.05. An adjusted *p*-value of less than 0.05 was considered statistically significant.

### 2.7. Ethics Statement

The procedures used in this study were in accordance with the Declaration of Helsinki and were approved by the Ethics Committee of Tsukazaki Hospital. Signed informed consent was obtained from all subjects after they were informed of the procedures. This study was registered as UMIN000034493: “Prediction of the refraction after cataract surgery using artificial intelligence.”

### 2.8. Data Availability

Due to the nature of this research, participants of this study did not agree for their data to be shared publicly, so supporting data is not available.

## 3. Results

As a result of the study using the feature importance of GBR, the SRK/T equation was selected as the conventional equation to be used as a candidate for explanatory variables. As a result of the subsequent grid search, axial length, radius of corneal curvature, anterior chamber depth, lens thickness, IOL power, and predicted value using the SRK/T formula were selected as explanatory variables. Therefore, we used a combination of these explanatory variables in all ML methods. The absolute prediction error of the conventional IOL power calculation formulae is depicted in Figure 3. Mean absolute error for the Barrett Universal II formula, SRK/T formula, Holladay 1 formula, Hoffer Q formula, and Haigis formula were 0.2960, 0.3314, 0.3312, 0.3602, and 0.3210, respectively. The p-value by the Friedman test was < 0.0001. The Barrett Universal II formula provided values that were significantly lower than those provided by other formulae in terms of absolute prediction error.

Since the prediction error of the Barrett Universal II formula was the lowest, the accuracy of ML methods was compared to that of the Barrett Universal II formula. Before ML, we selected the parameters that should be used in cross-validation in the training data. The selected parameters were the same combination in all ML methods, which included axial length, radius of corneal curvature, IOL power, lens thickness, anterior chamber depth, and predicted value using the SRK/T formula. These parameters were then used to conduct ML.

The absolute prediction error of the Barrett Universal II formula and ML methods is shown in Figure 4. Mean absolute error for the Barrett Universal II formula, SVR, GBR, RFR, and NN were 0.2960, 0.2877, 0.2929, 0.2964, and 0.2891, respectively. The *p*-value by the Friedman test was > 0.05. SVR and NN yielded lower absolute errors than the Barrett Universal II formula. RFR, GBR, and NN had lower median absolute errors than the Barrett Universal II. However, no significant difference in accuracy was observed among these groups.

Mean absolute error for the Barrett Universal II formula, SVR, GBR, RFR, and NN were 0.2960, 0.2877, 0.2929, 0.2964, and 0.2891, respectively. The p-value by the Friedman test was > 0.05.

Subgroup analysis based on axial length is shown in Figure 5, and no significant differences were observed among the axial length subgroups.

As the value predicted by the SRK/T formula was used as the explanatory variable, we compared absolute prediction error values provided by the SRK/T formula and ML methods (Figure 6). All ML methods yielded significantly lower absolute error values than did the SRK/T formula method.

The proportion of objects with absolute prediction errors less than 0.5 D is shown in Table 3. The ML methods had a higher proportion of objects than the Barrett Universal II formula. However, the values failed to reach statistical significance.

*p* values were calculated using chi-squared test. The correction of multiple comparisons was not performed.

Table 4 shows the results of the importance analysis for each factor of GBR, with the SRK/T formula playing the largest role.

## 4. Discussion

The current study compared the accuracy of conventional IOL power calculation formulas with that of ML-based methods in predicting postoperative refraction. The absolute prediction errors of the conventional formulas were nearly the same as those of previous reports [3,4,5]. In addition, the Barrett Universal II formula had low prediction error, which is in accordance with previous studies. Considering that the Barrett Universal II formula was the most accurate, it was then compared with the ML methods. The absolute mean prediction errors of SVR and NN were lower than that of the Barrett Universal II formula. However, no significant difference was observed. The ML method had a higher percentage of cases with an absolute prediction error of less than −0.5 D than the Barrett Universal II method. However, no significant differ-ence was found. However, no significant difference was observed.

As first described by Arthur Samuel in 1959, ML is a “Field of study that gives computers the ability to learn without being explicitly programmed”. ML methods are also often used in the field of ophthalmology. There are numerous reports about image identification using NN, and we have conducted several reports on this area as well [24,25]. The prediction of postoperative refraction values is suitable for ML because it calculates the numerical value (postoperative refraction) using a set of numerical values obtained with preoperative measurements.

In this study, the predicted postoperative refraction value obtained using the conventional IOL power calculation formula was used as an explanatory variable in ML. For actual learning, calculated values from the SRK/T formula were used, which resulted in significantly enhanced accuracy compared with the original SRK/T formula. As conventional IOL power calculation formulas can be used to calculate postoperative refraction with preoperative parameters based on optics, these parameters were considered more appropriate for incorporation into a means of prediction using the calculated value rather than each parameter independently.

Various types of IOLs were used in patients, and we used the data from these cases for training. This is because consecutive cases were used to eliminate bias, which probably not only resulted in higher volumes of training data but also contributed to an improvement in accuracy. Postoperative refraction is affected by the type of IOLs used; therefore, an IOL constant is applied in the conventional IOL power calculation formula for each lens. Conversely, in our model, the value predicted by the SRK/T formula was used as the explanatory variable, which also acted as the IOL constant. Since the A constant, optimized using training data, was used for calculations using the SRK/T formula, we think that the influence of IOL type on refraction can be corrected by using the value predicted using the SRK/T formula as the explanatory variable.

Recently, Sramka et al. have reported about ML methods that can predict refractions after cataract surgeries [16]. Moreover, they have investigated the accuracy of SVR and NN. Their NN model adopted the ensemble model. In contrast, the NN model in this study was simple and did not use an ensemble, and it was like the Hill-radial basis function (RBF) calculator [7]. The accuracy of the models was extremely like that of ours. The structure of the Hill RBF calculator has not been published, and therefore, discussing the relationship between network structure and prediction accuracy is difficult. However, refractive power prediction after cataract surgery uses relatively few parameters and is similar to NN; thus, a complicated network may not be necessary.

In the present study, we investigated the accuracy of four representative ML methods in the prediction of refractive power after cataract surgery. In terms of the mean absolute error value or the proportion of absolute prediction errors less than 0.5 D, SVR and NN were better than the other methods. However, no statistically significant difference was observed. Model selection is a major theme in ML [26,27], and several cheat sheets are available [28,29]. However, the accuracy of the model depends on the data set; thus, we must assess which model is superior in the data set. To date, it is unclear which of the four models assessed in this study is the best. However, more samples must be used to identify the most accurate model.

The present study had several limitations. First, the accuracy of our method was not compared with that of the Hill-RBF calculator (The RBF Calculator Physician Team, Haag-Streit Switzerland (Koeniz, Switzerland) and Mathworks (Natick, USA)) [17], which uses the ML method. The Hill-RBF calculator was primarily established for LENSTAR (Haag-Streit, Koeniz, Switzerland), and this was the primary reason for the lack of comparison. In addition, such a method is only applicable when the target refraction is −2.5 D or higher. The objects in this study included eyes with high myopia in which the target refraction is less than −3.0 D, and they are out of bounds for this calculator. Since no study has shown that the Hill-RBF calculator is significantly more accurate than the Barrett Universal II formula, we believe that, at the very least, our method is not inferior to the Hill-RBF calculator. The second problem is that we did not compare our formula with Kane’s one, because we emphasized the fact that many papers have shown that the Barrett Universal II formula is the best [5,6,7]. However, in terms of the number of cases studied, it is reasonable to point out that Kane’s formula is the most accurate, [8] and further studies to compare our formula with Kane’s one are needed.

Because the study was conducted in Japanese patients, who have very few short ocular axial patients, only a few cases with a short ocular axial length of less than 22 mm were included. Future studies are needed, such as using data from other races. Alternatively, the newly proposed formula using AS-OCT by Satou et al. that is independent of ocular axial length may be optimal [4]. However, it is complicated to obtain detailed AS-OCT data accurately from all preoperative cataract patients.

Another potential limitation of the present study is that some values used in our model, such as subjective refraction or IOL constants in power calculation formulae, can vary depending on the facility, suggesting that a model trained at one facility cannot be used with data from other facilities. Two methods to solve this problem are possible—namely, (i) a method of constructing a model based on large data from multiple facilities, and (ii) a method of constructing a model for each facility. Conceivably, facility-specific models would fit each facility better, albeit with smaller data volume. However, determining which method is more accurate is necessary. Furthermore, our method does not have a sufficient interface; therefore, the interface must be improved before its application in other facilities. At present, the method is rather complicated, but once it reaches the implementation stage, learning and inputting can be automated, which will enable operation at the user level with the same level of work as the conventional method.

In conclusion, the predictive accuracy of the four ML methods for refractive power after cataract surgery was compared with that of the conventional IOL power calculation formulas. The accuracy of the ML models was not inferior to that of the Barrett Universal II formula, which is the most accurate among the conventional formulas. Furthermore, we obtained higher accuracy using the prediction result of the conventional formula as an explanatory variable than that using the original formula. This suggests the possibility of improving the accuracy of conventional formulae based on optical calculations. In general, the more training data ML has, the more accurate it becomes, so if the number of data is sufficient, the ML method can be applied to cases of abnormally shaped eyeballs. The results suggest that it may be possible to create a formula that is optimal for each facility on a facility-by-facility basis. Moreover, the method will not be affected by differences in race, and it can become the mainstream method for IOL power calculation in the future.

## Figures and Tables

**Figure 1 jcm-10-01103-f001:**
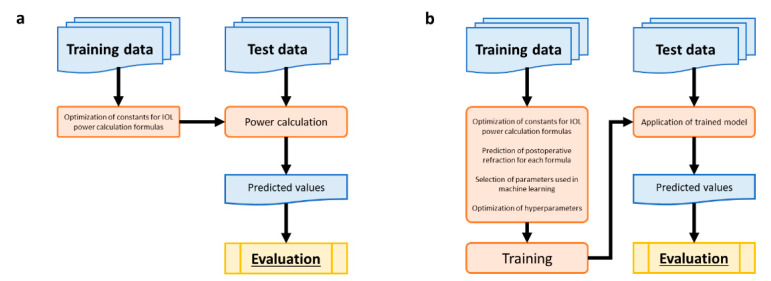
(**a**) Flow of prediction using the conventional intraocular lens (IOL) calculation formulas. (**b**) Flow of prediction using machine learning methods.

**Figure 2 jcm-10-01103-f002:**
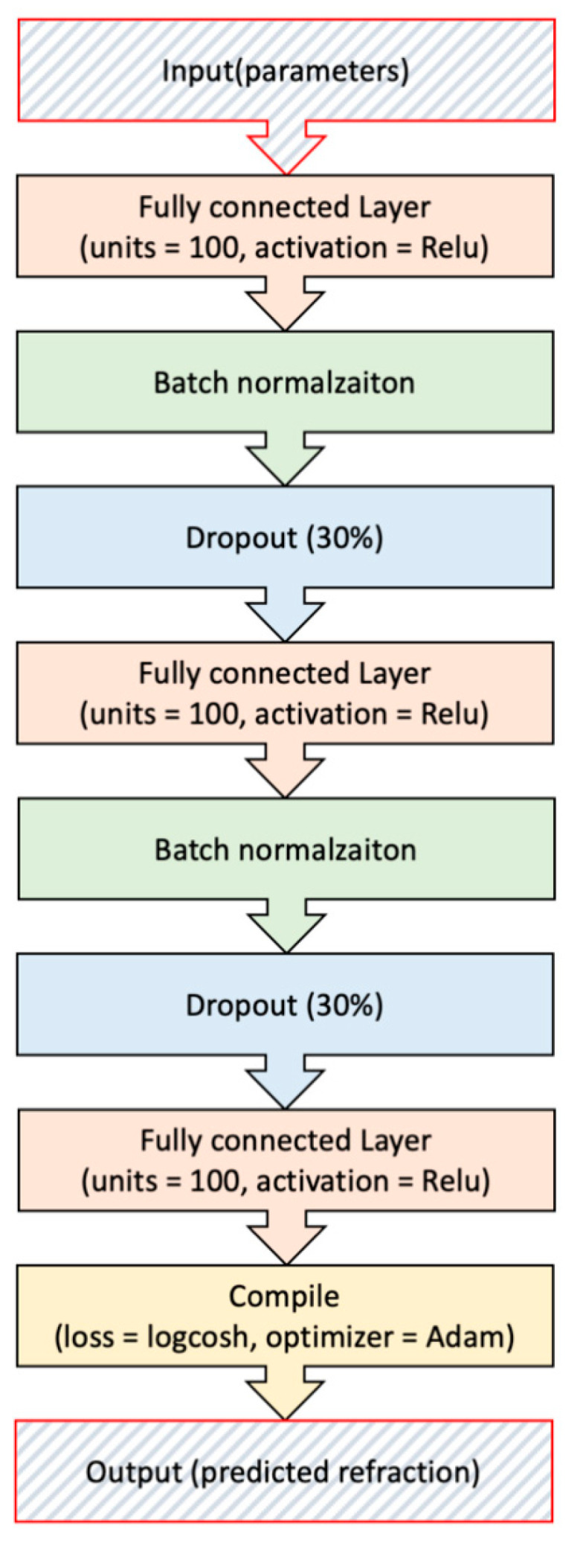
The structure of Neural Network (NN).

**Figure 3 jcm-10-01103-f003:**
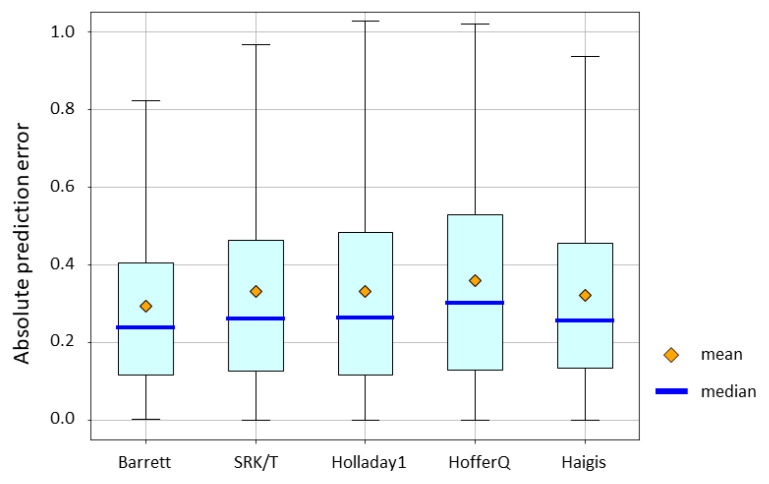
Mean absolute prediction error of conventional IOL power calculation formulae. Mean absolute error for the Barrett Universal II formula, SRK/T formula, Holladay 1 formula, Hoffer Q formula, and Haigis formula were 0.2960, 0.3314, 0.3312, 0.3602, and 0.3210, respectively. The *p*-value by the Friedman test was < 0.0001. The SRK/T formula vs. the Barrett Universal II formula: *p* = 0.0002. The Holladay 1 formula vs. the Barrett Universal II formula: *p* = 0.0004. The Hoffer Q formula vs. the Barrett Universal II formula: *p* < 0.0001. The Haigis formula vs. the Barrett Universal II formula: *p* = 0.0013. (The p-values were calculated using the Wilcoxon signed-rank test and were adjusted using the Bonferroni correction).

**Figure 4 jcm-10-01103-f004:**
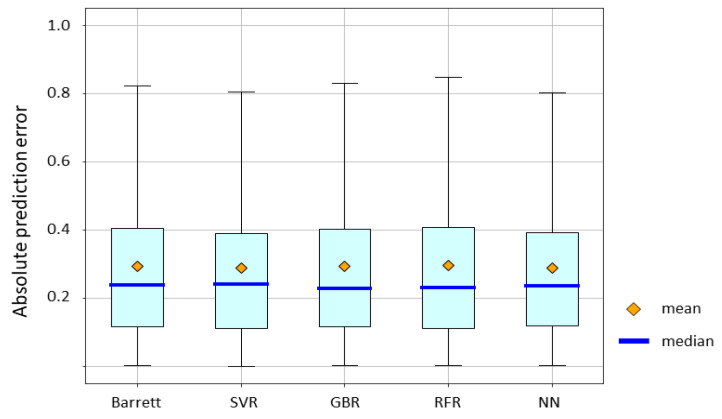
Mean absolute prediction error value of the Barret Universal II formula and machine learning methods. Abbreviations: SVR, support vector regression; GBR gradient boosting regression; RFR, random forest regression; NN, neural network.

**Figure 5 jcm-10-01103-f005:**
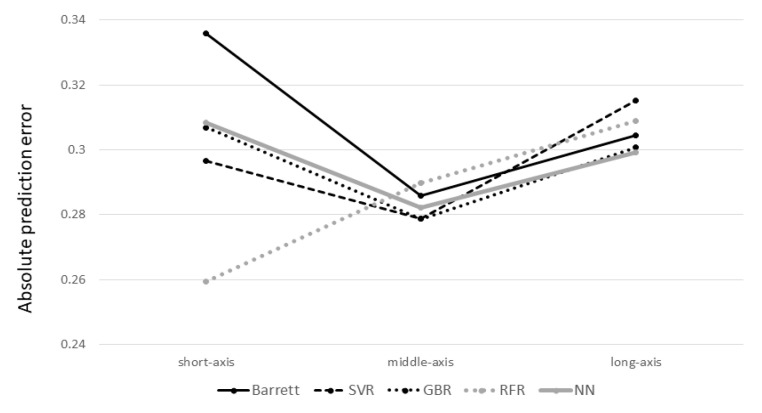
Mean absolute prediction error categorized according to the axial length. In the short-axis group, the mean absolute error for the Barrett Universal II formula, SVR, GBR, RFR, and NN were 0.3360, 0.2967, 0.3069, 0.2593, and 0.3085, respectively. In the middle-axis group, the mean absolute error for the Barrett Universal II formula, SVR, GBR, RFR, and NN were 0.2858, 0.2789, 0.2788, 0.2898, and 0.2821, respectively. In the long-axis group, the mean absolute error for the Barrett Universal II formula, SVR, GBR, RFR, and NN were 0.3045, 0.3153, 0.3008, 0.3089, and 0.2991, respectively. The *p*-value by the Friedman test was > 0.05 for all axial length subgroups.

**Figure 6 jcm-10-01103-f006:**
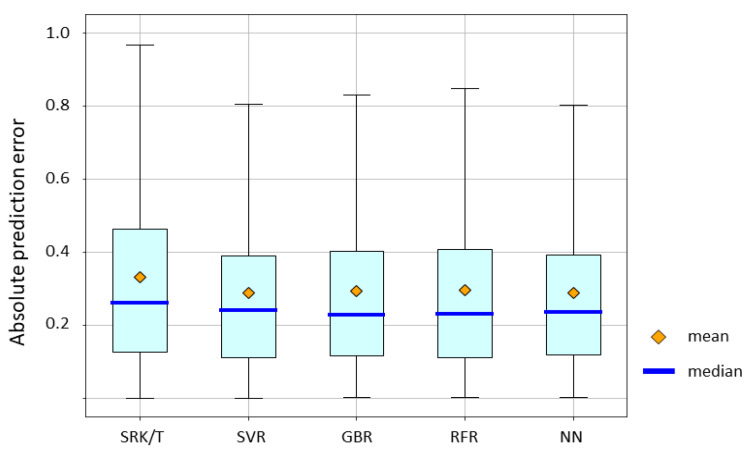
Mean absolute prediction error of the SRK/T formula and machine learning methods. Mean absolute error for the SRK/T formula, SVR, GBR, RFR, and NN were 0.3314, 0.2877, 0.2929, 0.2964, and 0.2891, respectively. The *p*-value by the Friedman test was < 0.0001. The SVR formula vs. the SRK/T formula: *p* < 0.0001; GBR vs. the SRK/T formula: *p* < 0.0001; RFR vs. the SRK/T formula: *p* = 0.0001; NN vs. the SRK/T formula: *p* < 0.0001; (The *p*-values were calculated using the Wilcoxon signed-rank test and were adjusted using the Bonferroni correction).

**Table 1 jcm-10-01103-t001:** Summary of the training and test data.

	Training Data	Test Data	*p* Value
*n*	Total: 2831 YP2.2: 296 SZ-1: 260 W60R: 193 KS-SP: 28 NS60YG: 21 SN60WF: 125 SN6AT: 208 SN6AD: 79 SV25T: 38 ZCB00V: 463 TECNIS multi: 463 TECNIS symphony: 202	Total: 500 YP2.2: 500	
Axial length	24.02 ± 1.57	23.92 ± 1.35	0.1741
Average radius of the corneal curvature	7.63 ± 0.27	7.62 ± 0.25	0.6757
ACD	3.10 ± 0.41	3.10 ± 0.38	0.9293
LT	4.57 ± 0.43	4.57 ± 0.43	0.5257
WTW	11.74 ± 0.41	11.75 ± 0.41	0.5651
IOL power	19.63 ± 4.25	19.55 ± 3.50	0.6942
Postoperative refraction	−0.13 ± 0.82	−0.09 ± 0.92	0.3323

Table 1 contains a summary of the training and test data. No significant difference was observed between the two groups in age, axial length, radius of corneal curvature, lens thickness, anterior chamber depth, white-to-white distance, refractive power of the implanted IOL, and postoperative refraction. Abbreviations: ACD, anterior chamber depth; LT, lens thickness; WTW, white-to-white distance *p*-values were calculated using an unpaired *t*-test. The correction of multiple comparisons was not performed.

**Table 2 jcm-10-01103-t002:** Optimized Constants for IOL power calculation formula.

A Constants (for SRK/T)		SF (for Holladay 1)		pACD (for Hoffer Q)		a0, a1, a2 (for Haigis)	
YP2.2	119.2	YP2.2	1.93	YP2.2	5.792	YP2.2	−1.72, 0.277, 0.260
SZ-1	119.48						
W60R	119.49						
KS-SP	119.72						
NS60YG	120.88						
SN60WF	119.2						
SN6AT	119.16						
SN6AD	119.24						
SV25T	119.56						
ZCB00V	119.58						
Tecnis multi	119.63						
Tecnis symphony	119.19						

These values were used as predicted refractions. Abbreviations: SF, surgent factor; pACD, personalized anterior chamber depth.

**Table 3 jcm-10-01103-t003:** Proportion of objects with errors less than 0.5 D.

Barrett Universal II Formula	SVR	RFR	GBR	NN
406/500	422/500	412/500	414/500	422/500

Abbreviations: SVR, support vector regressor; RFR, random forest regressor; GBR, gradient boosting regressor; NN, neural network. Barrett Universal II formula vs. SVR: *p* = 0.1800; Barrett Universal II formula vs. RFR: *p* = 0.6229; Barrett Universal II formula vs. GBR: *p* = 0.5102; Barrett Universal II formula vs. NN: *p* = 0.1800.

**Table 4 jcm-10-01103-t004:** Means of five cross validation for Characteristic Importance of GBR.

	Axial Length	Corneal Curvature	ACD	LT	IOL Power	SRKT
Mean	0.131356521	0.181638074	0.204091	0.173306	0.075068989	0.234539

Abbreviations: GBR, gradient boosting regressor; ACD, anterior chamber depth; LT, lens thickness.

## Data Availability

Due to the nature of this research, participants of this study did not agree for their data to be shared publicly, so supporting data is not available.

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
