# Peer review of "Use of a Machine Learning Method in Predicting Refraction after Cataract Surgery"

_jcm, 2021, doi:10.3390/jcm10051103_

Round 1

Reviewer 1 Report

Please move IOL to be spelled out earlier in the introduction. 

On page 2 of 13, Line 75 should state "We used a total of 2831 eyes", not 3331. 

2.4 surgical procedures can either be omitted except for objective measurements like incision size and rhexis size, or all steps of surgery should be described (viscoelastic removal, etc).

Please include city and country of software programs like python and Scikit.

Page 6 of 13, line 169, should give objective measurement for why gender was not included. Either state that it was found to be not statistically significant or provide actual numbers.

Author Response

#1. Please move IOL to be spelled out earlier in the introduction. 
Thank you. I spelled it out.

#2. On page 2 of 13, Line 75 should state, "We used a total of 2831 eyes", not 3331. 
Thank you for pointing this out. I corrected it.

#2. 2.4 surgical procedures can either be omitted except for objective measurements like incision size and this size, or all steps of surgery should be described (viscoelastic removal, etc).
Thank you for your comment.
As pointed out, I removed the surgical procedures section. I added “surgical measurements” to the previous section's title and connected the following sentence to the end of the previous section.
“Six experienced surgeons performed cataract surgeries. 2.4-mm temporal corneal incision was made. Next, a 5.0-mm continuous curvilinear capsulorhexis was created.”

#3. Please include city and country of software programs like python and Scikit.
Thank you for pointing this out. Each was added as follows. (Scikit is a library distributed on the net community, so it has no location.)
“Python 3 (https://www.python.org/ Python Software Foundation, Delaware, USA) and Scikit-learn library (http://scikit-learn.org/stable/ Free software machine learning library for the Python programming language)”

#4. Page 6 of 13, line 169, should give an objective measurement for why gender was not included. Either state that it was found to be not statistically significant or provide actual numbers.

Thank you for your good advice. Our unpublished analysis results, which is the basis for not considering gender, have been added with specific figures as follows.
“It showed that the feature importance of the parameters (axial length, corneal curvature radius, lens power, lens thickness, white to white, ACD, A constant, age, gender) was 217, 123, 121, 24, 23, 22, 21, 17, and 13, respectively, in F-value with the gender being the smallest.”

Reviewer 2 Report

The manuscript describes a study related to usage of ML methods for evaluating postoperative refraction after cataract surgery.

The topic is important and worth of research efforts, but I think the presented manuscript does not provide significant contributions.

First of all, the topic is not adequately introduced and motivated, and many important concepts about the domain knowledge are missing, in a way that the manuscript is very hard to understand.

Discussion of related work is incomplete: the following important papers were not cited and discussed

Satou, Tsukasa, et al. "Development of a new intraocular lens power calculation method based on lens position estimated with optical coherence tomography." Scientific reports 10.1 (2020): 1-11.

Kane, Jack X., and David F. Chang. "Intraocular Lens Power Formulas, Biometry, and Intraoperative Aberrometry: A Review." Ophthalmology (2020).

Gundersen, Kjell Gunnar, and Richard Potvin. "Comparing Visual Acuity, Low Contrast Acuity and Refractive Error After Implantation of a Low Cylinder Power Toric Intraocular Lens or a Non-Toric Intraocular Lens." Clinical Ophthalmology (Auckland, NZ) 14 (2020): 3661.

The processing pipeline is not clear: what kind of data is used for developing the models? How are formulas computed? What are the important parameters? what are the input data for the ML models?

Statistical analysis is not convincing: empirical formulas seem to have similar performances, and I sincerely don't understand the need of a more complicated ML model to substitute them.

It would be different if ML models are developed in a way that complicated measures do not need to be taken, but it does not seem the case, at least according to the manuscript.

Author Response

#1.
The manuscript describes a study related to usage of ML methods for evaluating postoperative refraction after cataract surgery. The topic is important and worth of research efforts, but I think the presented manuscript does not provide significant contributions. First of all, the topic is not adequately introduced and motivated, and many important concepts about the domain knowledge are missing, in a way that the manuscript is very hard to understand. Discussion of related work is incomplete: the following important papers were not cited and discussed
Satou, Tsukasa, et al. "Development of a new intraocular lens power calculation method based on lens position estimated with optical coherence tomography." Scientific reports 10.1 (2020): 1-11.
Kane, Jack X., and David F. Chang. "Intraocular Lens Power Formulas, Biometry, and Intraoperative Aberrometry: A Review." Ophthalmology (2020).
Gundersen, Kjell Gunnar, and Richard Potvin. "Comparing Visual Acuity, Low Contrast Acuity and Refractive Error After Implantation of a Low Cylinder Power Toric Intraocular Lens or a Non-Toric Intraocular Lens." Clinical Ophthalmology (Auckland, NZ) 14 (2020): 3661.

Thank you for your valuable advice.

I followed your advice and added Satou and Kane's paper to the Introduction and Discussion sections. Gundersen’s article is about Toric Lens. After careful consideration with the co-authors, it was decided that this paper, the subject of which is an intraocular lens calculation formula using artificial intelligence, was not suitable for citation.

*Satou’s paper
<Introduction>
“Satou et al. have recently reported a formula that uses detailed anatomical measurements of the anterior eye using anterior segment Optical Coherence Tomography (AS-OCT). Their formula shows high accuracy without being affected by the axial length of the eye.”
<Discussion>
“Alternatively, the newly proposed formula using AS-OCT by Satou et al. that is independent of ocular axial length may be optimal. However, it is complicated to obtain detailed AS-OCT data accurately from all preoperative cataract patients.”

*Kane’s Paper
<Introduction>
“On the other hand, Kane et al. recently reported that Kane's formula has higher accuracy in studies with many cases.”
<Discussion>
“The second problem is that we did not compare our formula with Kane's one because we emphasized the fact that many papers have shown that the Barrett Universal II formula is the best [5-7]. However, in terms of the number of cases studied, it is reasonable to point out that Kane's formula is the most accurate, [8] and further studies to compare our formula with Kane's one are needed.”

#2.
The processing pipeline is not clear: what kind of data is used for developing the models? How are formulas computed? What are the important parameters? what are the input data for the ML models?

Thank you for pointing out a significant omission.

I have added the following information to the beginning of the Result.

“As a result of the study using the feature importance of GBR, the SRK/T equation was selected as the conventional equation to be used as a candidate for explanatory variables. As a result of the subsequent grid search, axial length, radius of corneal curvature, anterior chamber depth, lens thickness, IOL power, and predicted value using the SRK/T formula were selected as explanatory variables. Therefore, we used a combination of these explanatory variables in all ML methods.”

#3.
Statistical analysis is not convincing: empirical formulas seem to have similar performances, and I sincerely don't understand the need for a more complicated ML model to substitute them.
It would be different if ML models are developed in a way that complicated measures do not need to be taken, but it does not seem the case, at least according to the manuscript.

Thank you for your crucial point.

As the reviewer pointed out, this method is complicated. Since this research is an experimental study, many parts are done manually, such as data forming and model creation, and ML is indeed a time-consuming method.

However, we can automate these tasks, and by automatically creating and applying models, we can ensure the same operability as conventional formulas at the user level.

The suggestion that this method is of little value because there is no difference in accuracy between it and the conventional method, our basic understanding is that it is of sufficient significance if it has the same level of accuracy as the conventional method with the highest accuracy. At present, many formulas for calculating intraocular lens power are in use, and it is known that each formula has its own specialties (eye shape), so I believe that increasing the number of options for calculation methods is valuable. Besides, although the data is unpublished, we have repeatedly conducted similar studies before. Although there are often no significant differences, we have confirmed that ML is basically more accurate than conventional formulas. Because the nature of ML is such that accuracy increases with the amount of training data, we believe that ML will become one of the main methods in the future.

Based on the comments, I added the following sentence to the Discussion.
“At present, the method is rather complicated, but once it reaches the implementation stage, learning and inputting can be automated, which will enable operation at the user level with the same level of work as the conventional method.”

Round 2

Reviewer 2 Report

All comments were addressed. The paper is ok now.

This manuscript is a resubmission of an earlier submission. The following is a list of the peer review reports and author responses from that submission.

Round 1

Reviewer 1 Report

Yamauchi et al present a retrospective series aiming to use machine learning for IOL power calculation, and comparing this to conventional IOL power calculation formula.

In its current form, the manuscript is not prepared properly. The sections are completely out of order, and there is evident text (Lines 49-64) that does not belong in the manuscript. In its current form, the manuscript is exceedingly difficult to follow.

I encourage the authors to review the manuscript and to ensure a copy that is formatted properly has been uploaded. At that time I will be happy to provide a review.

Author Response

I'm very sorry.
I'm attaching a revised version that addresses the other reviewer's comments as well as fixing the structure.
Thank you very much for your review.

Reviewer 2 Report

Yamauchi et al. describe the implementation of machine learning algorithms for postoperative error refraction prediction among surgically corrected cataract cases. The authors build upon prior publications by Roberts (2018), Kane (2017), and Shajari (2018), and report non-inferiority of these methods to the Barrett Universal II (BUII) formula as gold-standard with significant sample size. The proposal is technically innovative by addressing postoperative refraction error prediction with novel methodology among the Japanese patient population. However, some major and minor concerns need to be addressed before publication.

Major concerns

  1. The inclusion of intraocular lens calculation by BUII as an explanatory variable in the ML modeling might not be appropriate since there could be collinearity among BUII and several included predictors in the model.
  2. Baseline characteristics of the patient population, especially ocular history, and comorbidities should be reported. The authors should also describe if the analyzed population included patients with refraction comorbidities such as keratoconus. The latter has been reported that IOL calculation and postoperative error refraction is challenging and often imprecise given the corneal curvature inclination and greater anterior chamber depth.
  3. Lines 246-247. The authors should be mindful of political correctness when describing study participants. Patients, as described in these lines, are not "Objects."
  4. To improve existing IOL and postoperative refraction error calculations, why not include other eye biometric measurements? or patient characteristics? Even surgeon factors? Rather than replicating findings with a machine-learning approach.
  5. Lines 117-118. What was the importance of each variable in the ML model? Please include this information. It may be the case that ML is non-inferior to standard IOL postoperative refraction formulas because the model primarily relied on SRK/T results.
  6. Lines 246-147. How many eyes were analyzed in this proposal? All patients had both eyes intervened? Please be specific here in describing the units of analysis.
  7. Lines 284 and 292. Why did the authors conduct statistical analysis in Excel? The authors described their ML modeling with Python. This software should be preferred over Excel.

Minor concerns

  1. Lines 49 to 64 ("Experimental section") need to be erased.
  2. Figures 2 and 3 provide a clear depiction of data. However, the authors should also consider tabulating these results as supplementary information.

Author Response

Dear Reviewer 2

Thank you for taking time out of your busy schedule to review our paper.

We apologize for our mistakes that left extra sections in place.

We have revised the text in accordance with your valuable comments and would appreciate a re-review.

Major concerns

1 .The inclusion of intraocular lens calculation by BUII as an explanatory variable in the ML modeling might not be appropriate since there could be collinearity among BUII and several included predictors in the model.

Thank you for your thoughtful comments.

As mentioned in the Machine learning part of the Method, “We excluded the predicted value of the Barrett Universal II formula from the candidates for explanatory variables because the details of the formula have not been published. “we used four expressions, SRK/T formula, Holladay 1 formula, Hoffer Q formula, and Haigis formula as candidate explanatory variables and did not use BUII. In line with the intentions, you pointed out, I added "also to avoid autoregressive behavior" to the Method.

2.Baseline characteristics of the patient population, especially ocular history, and comorbidities should be reported. The authors should also describe if the analyzed population included patients with refraction comorbidities such as keratoconus. The latter has been reported that IOL calculation and postoperative error refraction is challenging and often imprecise given the corneal curvature inclination and greater anterior chamber depth.

Thank you for your careful remarks.

Basically, we exclude cases with eye diseases other than cataracts that may affect visual function. Cases that are not expected to significantly affect visual function, such as mild glaucoma or mild diabetic retinopathy, are included. Cases of keratoconus that could be diagnosed by an ophthalmologist are excluded.

The following was added to 2.2 the Patients section. " For example, Keratoconus, moderate or greater glaucoma or diabetic retinopathy were excluded " 

3.Lines 246-247. The authors should be mindful of political correctness when describing study participants. Patients, as described in these lines, are not "Objects."

We appreciate you pointing out our lack of consideration. We have rewritten Objects to Patients in 2.2 Patients section.

4.To improve existing IOL and postoperative refraction error calculations, why not include other eye biometric measurements? or patient characteristics? Even surgeon factors? Rather than replicating findings with a machine-learning approach.

Thank you for your comments on our lack of explanation.

The ones we used here are the ones commonly used in traditional formulas, such as axial length, corneal curvature, anterior chamber depth, lens thickness, and white-to-white distance. As for 'patient characteristics', we have included age in this case. Gender was not considered because it did not contribute much to accuracy in our previous unpublished study. We believe that 'surgeon factors', which are contributing factors for each facility and surgeon, are important. The reason for the occurrence of surgeon factors is mainly considered to be the variability of the pre- and post-operative examinations at each facility, but since all the operations were performed at the same facility in this study, we believe that the effect is small.

The following was added to 2.6 Machine learning section “Gender was not considered because it did not contribute much to accuracy in our previous unpublished study.”

5.Lines 117-118. What was the importance of each variable in the ML model? Please include this information. It may be the case that ML is non-inferior to standard IOL postoperative refraction formulas because the model primarily relied on SRK/T results.

We appreciate your comments that add to the depth of the paper.

We have indicated the importance of each factor of the gradient boost in the Results as you pointed out (Table 3), with the SRK/T equation resulting in the highest value. From this result, we think you are right, and we suspect that the SRK/T equation largely determines the predictions, and that the other explanatory variables increase the accuracy by fine-tuning the predictions.

6. Lines 246-147. How many eyes were analyzed in this proposal? All patients had both eyes intervened? Please be specific here in describing the units of analysis.

Thank you for your important comment.

First, since it is known that both eyes of the same patient are well correlated, the inclusion of the opposite eye of the test cases in the training cases may lead to a greater overestimation of the accuracy of ML than is actually the case. Therefore, we are testing 500 eyes, all with entries from one eye only.

We have added the following text to the Patients Section.

We used a total of 3331 eyes for training for ML: 487 unilateral entries in 487 cases and 1172 binocular entries in 2344 eyes.For the evaluation of ML performance, 500 eyes per model were tested using only unilocular entries.

7.Lines 284 and 292. Why did the authors conduct statistical analysis in Excel? The authors described their ML modeling with Python. This software should be preferred over Excel.

Thank you for your comment.

As mentioned in 2,7 Statistical analysis “For statistical analysis, we used Python 3 and the SciPy library (https://www.scipy.org/)”. we do not use Excel for the statistics; Excel is used to optimize the traditional intraocular lens constants.

Minor concerns

1. Lines 49 to 64 ("Experimental section") need to be erased. 

Thank you.

The entire section has been erased and the entire text organized.

2. Figures 2 and 3 provide a clear depiction of data. However, the authors should also consider tabulating these results as supplementary information.

Thank you for your advice.

Figure 2 and Figure 3 are moved to Supplemental Figure 2 and Supplemental Figure 3 respectively. However, in accordance with the editorial policy of this journal, we have inserted the figures in the order they appeared in the text.

Round 2

Reviewer 2 Report

The authors addressed all of my concerns. Many thanks to the authors for thorough consideration and rapid response to my comments.